# Characterization of the Observational Covariance Matrix of Hyper-Spectral Infrared Satellite Sensors Directly from Measured Earth Views

**DOI:** 10.3390/s20051492

**Published:** 2020-03-09

**Authors:** Carmine Serio, Guido Masiello, Pietro Mastro, David C. Tobin

**Affiliations:** 1School of Engineering, University of Basilicata, 85100 Potenza, Italy; carmine.serio@unibas.it (C.S.); pietro.mastro@unibas.it (P.M.); 2CIMSS/SSEC, University of Wisconsin, Madison, WI 53706, USA; dave.tobin@ssec.wisc.edu

**Keywords:** instrument radiometric characterization, radiometric noise, hyper-spectral sounders, infrared, satellite

## Abstract

The observational covariance matrix, whose diagonal square root is currently named radiometric noise, is one of the most important elements to characterize a given instrument. It determines the precision of measurements and their possible spectral inter-correlation. The characterization of this matrix is currently performed with blackbody targets of known temperature and is, therefore, an output of the calibration unit of the instrument system. We developed a methodology that can estimate the observational covariance matrix directly from calibrated Earth-scene observations. The technique can complement the usual analysis based on onboard blackbody calibration and is, therefore, a useful back up to check the overall quality of the calibration unit. The methodology was exemplified by application to three satellite Fourier transform spectrometers: IASI (Infrared Atmospheric Sounder Interferometer), CrIS (Cross-Track Infrared Sounder), and HIRAS (Hyperspectral Infrared Atmospheric Sounder). It was shown that these three instruments are working as expected based on the pre-flight and in-flight characterization of the radiometric noise. However, for all instruments, the analysis of the covariance matrix reveals extra correlation among channels, especially in the short wave spectral regions.

## 1. Introduction

The characterization of the radiometric noise, or more in general the observational covariance matrix, of remote sensing instrumentation is an important step to fully exploit the related observations for the estimation of geophysical parameters. The observational covariance matrix, $S_{\varepsilon }$, is an ingredient of the so-called Level 1 (L1) data. For infrared sensors, such as those we deal with in this paper, L1 data correspond to the calibrated spectral radiance. The diagonal of $S_{\varepsilon }$ corresponds to the variances of the observations and their square root is what is normally referred to as the radiometric noise. This expresses the precision of measurements because of instrument noise. The off-diagonal terms of $S_{\varepsilon }$ (normally termed as covariances) give the inter-correlation among observations. Generally, off-diagonal terms are negligible; however, non-zero covariances can arise because of post-elaboration of L1 data, such as mathematical apodization (e.g., see [1]), but they can also reflect unexpected noise sources, such as mechanical micro-vibrations, thermal effects and so on [2,3,4].

The radiometric noise of a given instrument is normally assessed through the calibration unit with the instrument looking at blackbody targets of known temperature. By accumulating many blackbody observations, one obtains an ensemble of spectra from which $S_{\varepsilon }$ is calculated considering ensemble variances and covariances. However, once in orbit, a given instrument is looking at Earth-scene views, which do not have temperature such as that of the blackbody source. In addition, radiation is absorbed by the atmosphere in a way which depends on the wave-number. The integrated radiation at the top-of-atmosphere can be thought of emission from a continuum range of temperature. Therefore, it is desirable to check if there are noise-dependencies on the Earth views.

Towards this objective, we devised a methodology that directly uses the spectral radiance observed from the instrument. In practice, the technique computes the spectral residual Observations-Calculation (*O-C*) from a set of observed spectra, and, as in the case of blackbody spectra, $S_{\varepsilon }$ is calculated considering ensemble variances and covariances of the spectral residuals. In principle, the *Calculation* can be obtained in many ways. One obvious way is to use a forward model (e.g., [5]) with the atmospheric state vector obtained through a fitting procedure (e.g., [6]). This approach was used by Serio et al. [3], but it has the potential disadvantage of relying on a forward model, which could introduce a new source of errors, e.g., extra bias and/or variability.

Another approach consists in first projecting the data into an orthogonal basis; back-projecting the data to the physical space through a suitable *truncated* expansion, which can be thought of as a reconstruction or representation of the *signal or S*; and, finally, obtaining the spectral residual according to *O-S*, where now *S* plays the role of *Calculation*. The orthogonal basis could be the usual Fourier transform or, as an alternative, the Principal Component Analysis (PCA) [7]. Both techniques have been extensively used for dimensionality reduction of data (e.g., [8,9,10,11,12,13,14,15]), whereas the use of PCA to assess the noise performance of infrared instrument has been used for the American CrIS (e.g., see [2]), although in that study the tool was applied mostly to blackbody spectra. These, depending on a single temperature parameter, can be fitted with zero or at most one PC score. In [2], the concept of instrument noise as the additive composition of a correlated term and a truly random part was also introduced. They also suggest that using PCA, one can estimate the truly random component alone. This approach was also followed by Lee et al. (2019) [16] who analyzed Hyperspectral Infrared Atmospheric Sounder (HIRAS) data. However, we think that this additive model is useless and does not yield a correct representation of the instrument noise, which can show spectral correlation because of pre- and post-processing of calibrated radiances. As an example, IASI (Infrared Atmospheric Sounder Interferometer) is Gaussian apodized (e.g., see [1]), therefore the instrument noise affecting IASI spectral radiances is spectrally correlated. Quite recently, Han et al. (2015) [17] showed that the self-apodization corrections of CrIS (Cross track Infrared Sounder) spectral radiance can correlate the instrument noise.

This paper aims at showing that, once properly applied, the PCA approach can estimate the total instrument noise with its spectral correlation, if any, and therefore there is no need to introduce an additive noise model.

The application of PCA to direct observations of the Earth emission is complicated because the Earth–atmosphere system introduces a signal in the spectral radiance, which needs more PC scores to be fitted. The problem of how many PC scores to use is the biggest issue in applying PCA machinery. The problem was solved in a theoretical Bayesian Information context by Serio et al. [4], who applied an optimal criterion (the Bayesian Information Criterion (BIC) [18]) to select the suitable number of PC scores to represent the signal. In this way, there is no subjective user intervention within the scheme. The approach can directly determine the optimal number of PC scores to represent the signal, hence how to recover *S* and get the spectral residual *O-S* from which we can estimate $S_{\varepsilon }$.

This approach was considered to assess the instrument noise of IASI-C, which was launched on 7 November 2018 onboard the Metop-C payload (Metop stands for Meteorological Operational satellite). The first IASI (IASI-A) was launched in 2006; IASI-C is the third of the series.

Application of the technique was extended also to the American CrIS on board the United States Suomi National Polar Partnership (NPP) Polar-Orbiting Operational Environmental Satellite (e.g., see https://www.nasa.gov/mission_pages/NPP/news/cris-operational.html) and the Chinese HIRAS (Hyperspectral Infrared Atmospheric Sounder) flying on the payload FY-3D (see, e.g., http://gsics.nsmc.org.cn/data/en/code/FY3D.html).

The paper is organized as follows. Section 2 deals with a description of the data. A brief account of the methodology is also presented in this section. Section 3 is devoted to the presentation of results. Section 4 shows a discussion about the results.

## 2. Data and Methods

In this section, we first provide a discussion about the data we used and then show the basic of PCA-BIC methodology for the estimation of instrument noise: both radiometric noise and covariance matrix.

### 2.1. Data

The instrument IASI [19] was designed, developed, and manufactured by CNES (Centre National d’Etudes Spatiales) in France, within a joint-venture with the European Organization for the Exploitation of Meteorological Satellite (EUMETSAT). IASI is flying onboard the Metop (Meteorological Operational Satellite) and its main mission is to provide information on temperature and water vapor profiles. The spectral coverage of IASI extends from 645 to 2760 cm${}^{-1}$. The sampling interval is Δσ=0.25 cm${}^{-1}$, which, considering the bandwidth of 2115 cm${}^{-1}$, yields 8461 channels or spectral radiances for each single spectrum. The main characteristics of the IASI instrument are summarized in Table 1.

The data we used in this analysis were acquired by the IASI-C instrument, which is flying on Metop-C. We used a set of IASI spectra from one whole orbit on 21 November 2019. The data consist of spectral radiances at the level 1C. These are calibrated spectra and Gaussian apodized [1]. Because of Gaussian apodization, IASI level 1C radiances are correlated. The correlation introduced by apodization is shift-invariant, i.e. it does not depend on the wave number and, for a given channel centered at $\sigma_{0}$, it drops symmetrically to zero after three wave numbers. That is the correlation with a channel centered at generic wave number σ is zero if $∥\sigma -\sigma_{0}∥>3\times \Delta \sigma$, with Δσ=0.25 cm${}^{-1}$ the IASI sampling interval.

In effect, this is not the only source of correlation in IASI level 1C data. An extensive assessment of the IASI instrument noise performed on the basis of IASI-A-B spectra has shown that IASI has extra correlation mostly affecting the spectral region around the merging of IASI bands [3,15]. This extra-correlation is likely the effect of mechanical micro-vibrations of the instrument, although data post-processing could also play a role.

The radiometric noise of IASI-C is shown in Figure 1 in terms of Noise Equivalent Difference radiance (NEDN). This is the square root of the diagonal of the covariance matrix shown in Figure 2. The observational covariance matrix, shown in Figure 2, also referred to as level 1C noise, applies to IASI apodized radiances. It is the best estimate as obtained by CNES engineers during the pre-flight and the first month in-orbit operations for IASI-C [20]. These noise figures have to be intended as an average for the four IASI IFOVs or pixels.

Slight variations among pixels are expected to exist, although these are not a concern of this study, which mostly focused on exemplifying and demonstrating the PCA methodology.

In Figure 2, it is quite clear that there is an excess of correlation corresponding to spectral range centered at 1200 cm${}^{-1}$; an excess of correlation, although fainter, is also seen at 2000 cm${}^{-1}$. These are the spectral intervals where IASI bands 1 and 2 and bands 2 and 3 are merged, respectively. At both merging of bands, we have a long-range correlation structure with a cyclic behavior, which extends back to the beginning of band 1 at 645 cm${}^{-1}$. We stress that this behavior has been long ignored and not modeled for the covariance matrix of both IASI-A and -B. The structure was first evidenced by Serio et al. [3], Masiello et al. [15], who assessed that the correlation was not a result of atmospheric signal. We also stress that. for the correlation matrix shown in Figure 2, the covariance model has to be intended as an average over the four IASI pixels, and pixel-to-pixel variations can be expected. Finally, we note that the IASI-C noise model is still under analysis at CNES and refinements could be expected, although not changing the pattern shown in Figure 2. The radiometric noise and related covariance matrix in Figure 1 and Figure 2 are referred to as IASI *nominal noise* in the rest of this paper.

The American Cross-track Infrared Sounder (CrIS) provides infrared emission spectra over three wavelength ranges: LWIR (648.75–1096.25 cm${}^{-1}$), MWIR (1207.50–1752.50 cm${}^{-1}$), and SWIR (2150–2555 cm${}^{-1}$). As with IASI, CrIS has a calibration unit, which performs views of the internal calibration target (ICT, warm calibration point) and a deep space view (DS, cold calibration point).

The three spectral bands of CrIS are not merged together and their spectral sampling can depend on the way the interferogram is sampled. In fact, the instrument can be operated in the full spectral resolution (FSR) mode, which corresponds to the interferogram recorded with the maximum optical path difference (MOPD) equal to 0.8 cm for each band. In this case, the sampling is Δσ=0.625 cm${}^{-1}$ for all bands. However, for operations, the CrIS interferogram can also be recorded in the so-called normal spectral resolution mode (NSR) with MPOD of 0.8, 0.4, and 0.2 cm for the LWIR, MWIR, and SWIR bands, respectively, which leads to the sampling of Δσ=0.625 cm${}^{-1}$ for the LWIR, 1.25 cm${}^{-1}$ for the MWIR, and 2.5 cm${}^{-1}$ for the SWIR. The main characteristics of the CrIS instrument are summarized in Table 2.

The data we used for CrIS were acquired on 30th of September 2015 and 9th of November 2015. The CrIs was operated in the normal spectral resolution or NSR mode. The observations consist of unapodized spectral radiances at the normal spectral resolution of 0.625 1.25, and 2.5 cm${}^{-1}$ for bands LWIR, MWIR, and SWIR, respectively.

Figure 3 shows the radiometric noise for CrIS in terms of NEDN (e.g., [17]). The data shown in Figure 3 have been reprocessed because the original radiometric noise refers to unapodized spectral radiances at the FSR mode of 0.625 cm${}^{-1}$ for all CrIS bands. The data points for band 2 (MWIR) have been reduced a factor $\sqrt{2}$ and those for band 3 (SWIR) a factor 2 to take into account the resampling at 1.25 and 2.5 cm${}^{-1}$, respectively. Considering the work by Han et al. (2015) [17], this should be kept in mind when comparing to our results that we show in Section 3. The radiometric noise figures in Figure 3 are referred to as CrIS *nominal noise* in the remaining of this paper. Unlike IASI, for CrIS, there is no special prescription for the off-diagonal terms of the covariance matrix, which is normally assumed diagonal. This applies indifferently to unapodized radiances in the FSR mode or NSR mode.

The Chinese HIRAS (Hyperspectral Infrared Atmospheric Sounder, e.g., [21]) is quite similar to CrIS as far as the instrument concept is concerned and similar to IASI for the scan pattern geometry (29 FOR covering a swath of 2800 km; each FOR consists of a 2×2 matrix of IFOVs; and each IFOV has a ground resolution of 16 km). HIRAS has three spectral bands: LWIR or band 1 (650–1136 cm${}^{-1}$), MWIR or band 2 (1210–1750 cm${}^{-1}$), and SWIR or band 3 (2155–2550 cm${}^{-1}$). As with CrIS, the full spectral resolution is 0.625 cm${}^{-1}$ and each HIRAS scan includes views of the internal calibration target (ICT, warm calibration point) and a deep space view (DS, cold calibration point). The spectra acquired during these views are used to estimate the radiometric noise, which consists of spectral standard deviation of HIRAS spectra acquired by looking at the cold space and at the internal warm blackbody target. The warm and cold targets’ noise estimates are provided with the HIRAS data. For our purpose, they were averaged to form the HIRAS *nominal noise*, which is shown in Figure 4. As done for CrIS and IASI, for HIRAS, the nominal noise in Figure 4 was obtained by averaging over the four HIRAS pixels.

For HIRAS, we have a dataset acquired on 15 July 2019 over the Indian Ocean. The data consist of unapodized, calibrated spectral radiances at FSR of 0.625 cm${}^{-1}$. Similar to CrIS, for HIRAS, there is no special prescription for the off-diagonal terms of the covariance matrix, which is assumed to be diagonal.

### 2.2. Methods

In this section, we summarize the PCA methodology we devised to get an estimate of the observational covariance matrix of high spectral resolution infrared sensors.

First, let R be a radiance vector of size *d*. We assume to have an ensemble of observed *d*-dimensional vectors $R_{i},\;i=1,\ldots ,N$. The ensemble average $\bar{R}$ is defined as usual,
(1)$$\bar{R}=\frac{1}{N}\sum_{i=1}^{N}R_{i}$$
For the generic radiance vector R, we assume an additive signal noise model,
(2)R=s+ε
with s the signal and where the noise term ε is assumed Gaussian, zero mean, and covariance matrix $S_{\varepsilon }$. Next, we define the normalized, zero mean, vector, $x_{i}$, according to
(3)$$x_{i}={\tilde{S}}_{\varepsilon }^{-\frac{1}{2}}\left(R_{i}-\bar{R}\right)$$
where ${\tilde{S}}_{\varepsilon }$ is any suitable a priori estimate of the observational covariance matrix. This should not be confused with the true, unknown, noise covariance, $S_{\varepsilon }$. The normalizing covariance ${\tilde{S}}_{\varepsilon }$ can be computed in many ways, e.g., it could be set to the radiometric specifications or it could be assumed equal to any estimation obtained during the pre-flight operations. If available, one could use the best estimate obtained, e.g., based on the blackbody calibration unit.

Let X be the d×N matrix whose columns are the normalized observations, $x_{i}$. An orthogonal basis for the *d*-dimensional vectors $x_{i}$ is obtained by Singular Value Decomposition (SVD) of the symmetric, covariance matrix
(4)$$S=\frac{1}{N}XX^{t}$$
where the superscript *t* means transpose operation. Finally, the SVD decomposition of S is obtained according to
(5)$$S=U\Lambda U^{t}$$
with the matrix U unitary and orthogonal and where Λ is a diagonal with elements (eigenvalues), $\lambda_{j},\;j=1,\ldots ,d$, and with $\lambda_{1}>\lambda_{2}>\ldots \lambda_{d}$. The PCA scores are given by,
(6)$$c_{i}=U^{t}x_{i}$$
and the elements of the vector $c_{i}$ are uncorrelated, i.e. their covariance matrix is diagonal values $\lambda_{j}$, j=1,…,d. The matrix U has size d×d and, considering only its first τ column (eigenvectors), we obtain a truncated principal component projection, or reconstructed radiance vector,
(7)$${\hat{R}}_{i}={\tilde{S}}_{\varepsilon }^{\frac{1}{2}}\left(U_{\tau }c_{i}\right)+\bar{R}$$
where the matrix $U_{\tau }$ is the matrix U with the first τ columns retained and the remaining d−τ set to zero. The truncation point τ can take a value within the range 1,…,d.

For a redundant signal, such as the spectral radiance, assuming correct choice of τ, an estimate, ${\hat{S}}_{\varepsilon }$ of the the covariance matrix, $S_{\varepsilon }$ of the noise term, ε can be obtained according to,
(8)$${\hat{S}}_{\varepsilon }=\frac{1}{N}\sum_{i=1}^{N}\left(R_{i}-{\hat{R}}_{i}\right){\left(R_{i}-{\hat{R}}_{i}\right)}^{t}$$
Considering Equation (Equation 7) and the orthogonal properties of the PC scores, after a bit algebra, we have
(9)$${\hat{S}}_{\varepsilon }={\tilde{S}}_{\varepsilon }^{\frac{1}{2}}U_{-\tau }\Lambda U_{-\tau }^{t}{\tilde{S}}_{\varepsilon }^{\frac{t}{2}}$$
with $U_{-\tau }=U-U_{\tau }$. Equation (Equation 9) has a simple meaning once we consider that the matrix $U_{-\tau }$ is the complement to $U_{\tau }$, that is it is the matrix U with the first τ columns zeroed and the remaining d−τ unchanged. In Equation (Equation 9), if we assume τ correct, the eigenvectors and eigenvalues of the signal are discarded, while only those corresponding to the noise are retained.

From Equation (Equation 9), it is seen that, if we change τ, we do not need to recompute the residuals. Furthermore, Equation (Equation 9) should not be confused with the covariance matrix of the reconstructed vector ${\hat{R}}_{i}$. The elements of the vector ${\hat{R}}_{i}$ are linearly dependent, because they have been obtained from τ<d independent PC scores, therefore the covariance matrix of ${\hat{R}}_{i}$ is singular: it does not have an inverse [15].

It has to be stressed that the correct choice of τ is critical for the scheme to work (e.g., see [4]). However, the SVD machinery does not contain any tool to choose the correct τ. If we take τ too small, we still have the contribution from the signal. If we take τ too large, we could reduce the bandwidth of the noise, which, we stress, extends over the total range of τ=1,…,d and underestimate the noise variance.

At a naive level, we might choose τ large enough that the signal has dropped to zero. However, we stress that *a too large τ reduces the noise bandwidth within the residuals* and should be avoided. One possible way to move on is to select τ based on a suitable optimal criterion, which guarantees that the main features of the signal are extracted. Such a criterion has been identified to be the Bayesian Information Criterion (BIC) [18]. For the problem at hand, it has been shown (e.g., [4]) that the BIC curve as a function of τ can be written as
(10)$$\mathrm{BIC}\left(\tau \right)=\left(N\sum_{j=1}^{d}\log\lambda_{j}+N\left(d-\tau \right)\log\left(\frac{1}{d-\tau }\sum_{j=\tau +1}^{d}\lambda_{j}\right)+\left(\tau +k\right)\log N\right)$$
with k=dτ−τ(τ−1)/2+d+1 [4]. One useful property of BIC is that it is guaranteed to select the true model, in case it is among the candidates, as N→∞. According to the BIC criterion, the optimal τ is that τ which minimizes Equation (Equation 10).

Before passing to results, we discuss the problem of how we can assess the statistical uncertainty of the estimator given by Equation (Equation 8). In effect, for a covariance statistics or estimator such as that of Equation (Equation 8), under the assumptions of Gaussian noise, the uncertainty is given by the Wishart distribution (e.g., [22]), which provides the statistical probability distribution of the sample covariance matrix.

Let $s_{ij}$ denote the true value of the element i,j of the matrix $S_{\varepsilon }$, then the variance of the estimate ${\hat{S}}_{\varepsilon }\left(i,j\right)$ is given by
(11)$$\mathrm{var}\left({\hat{S}}_{\varepsilon }\left(i,j\right)\right)=\frac{1}{N}\left(s_{ij}^{2}+s_{ii}s_{jj}\right)$$
which for i=j (variance or diagonal terms) becomes
(12)$$\mathrm{var}\left({\hat{S}}_{\varepsilon }\left(i,i\right)\right)=\frac{2}{N}\left(s_{ii}^{2}\right)$$
Thus, we see that the variance of the estimate is proportional to square of the true variance and inverse proportional to the statistical weight *N* or number of samples (spectra in our case). Therefore, the uncertainty of the radiometric noise estimate is, in percentage of the true value, of the order of ≈$\sqrt{2/N}$, which for the analysis presented here, considering that $N\approx 10^{4}$, yields a value of the order of ≈1–2%. As far as the bias is concerned, an extensive error analysis performed in simulation (e.g., see [4]) showed that the bias is nearly zero once the correct τ is selected, which is possible to do by using the BIC criterion (e.g., see [4]).

## 3. Results

In this section, we show the main results we have found by applying the methodology in Section 2.2 to IASI, CrIS, and HIRAS.

### 3.1. IASI

To implement the procedure for IASI, we need to select a proper ${\tilde{S}}_{\varepsilon }$ for normalization. This is assumed to be the very first CNES release (2006) of the covariance matrix, which was developed to characterize the instrument noise of IASI-A (e.g., see [4,23]). This matrix models only the correlation because of apodization. Additional source due, e.g., to mechanical vibrations are not represented. The radiometric noise corresponding to this matrix (that is the square root of the diagonal) is shown in Figure 5 and is compared with the best estimate of the nominal radiometric noise for IASI-C also shown in Figure 1. We purposely do not use the covariance matrix shown in Figure 2 for normalization, because we keep it for comparison to our results.

The results for the radiometric noise (square root of the diagonal of the matrix ${\hat{S}}_{\varepsilon }$ computed according to Equation (Equation 9)) are shown in Figure 6 for three different sets of observations:Cloudy tropical set: This consists of 14,321 spectra selected in the tropical belt (−35° to 35° latitude) with a cloud fraction CF=100%.Clear tropical set: This consists of 9324 spectra selected in the tropical belt (−35° to 35° latitude) with a cloud fraction CF≤5%.Clear High-Latitude set: This consists of 11,230 spectra selected at latitude (north and south) higher than 60° with a cloud fraction CF≤5%.

The cloud fraction was assessed with the native cloud mask of IASI Level 1C data as released by EUMETSAT (the cloud mask is based on the Advanced Very High Resolution Radiometer (AVHRR) imagery) and a cloud detection scheme developed in [24,25].

These three ensembles were set up to consider hot, warm, and cold Earth-scene views, hence to investigate a possible dependence of instrument noise on the scene radiation (scene photon noise or shot noise).

The results obtained by applying our PCA-BIC methodology are shown in Figure 6. A summary of the number of PC scores and τ selected by the BIC criterion is given in Table 3. Figure 6 shows that the methodology is capable of retrieving the correct radiometric noise also in the case the covariance matrix used for normalization is far apart from the true noise. We recall that, in the present analysis, the true radiometric noise was assumed to be that released by Chinaud et al. [20]. Figure 6 also shows that the estimation problem at hand is largely linear and our approach leads to convergence in one step (for further details, we refer the reader to [4]). It is also important to note that the radiometric noise does not show a strong dependence with the Earth-scene views. In effect, the tropical and High-Lat IASI spectra lead to almost exactly the same radiometric noise. In addition, it should be stressed that the radiometric noise released by CNES was assessed with blackbody spectra at a target temperature of some 290 K. The noise does not seem to be dominated by the scene radiation, a result which parallels that found for CrIS [2].

We also investigated a possible dependence of the radiometric noise on the pixel or IFOV. We remember that IASI has a Field of Regard (FOR) composed with a matrix of 2×2 pixels. These pixels or IFOVs are sensed through diverse detectors (the detectors are twelve because for each pixel we have three detectors, one for each of the three bands).

For the sake of brevity, the analysis is shown in Figure 7a for a set of all-sky tropical spectra (0≤CF≤100%). In addition, for this numerical experiment, the optimal τ selected through the BIC criterion is shown in Table 3. Figure 7a shows that there is no important dependence on the pixel. In the core of IASI band 1, it seems that pixels 1 and 2 perform slightly better than pixels 3 and 4. This results is also confirmed when we look at Figure 7b, where the CNES release is shown.

Finally, we investigated the problem of correlation among channels by considering the full estimated covariance matrix, ${\hat{S}}_{\varepsilon }$. In addition, in this case, for the sake of brevity, the analysis is shown for the set of tropical spectra with CF=100%. Figure 8 shows the full estimated observational matrix transformed to correlation. This can be compared to that released by CNES (Figure 2). Although the estimate shown in Figure 8 appears a bit noisy, the comparison with Figure 2 does show an excess of correlation especially in the merging of bands 1 and 2. The correlation structure around 1200 cm${}^{-1}$ is correctly reproduced, as it is possible to better see from Figure 9, which shows a zoom in the spectral interval where band 1 and 2 are merged. The comparison with the CNES release shows that the estimate obtained based on Earth-scene views is strikingly good in recovering the correlation patterns.

### 3.2. CrIS

For CrIS, the covariance matrix used to normalize the data is diagonal with the diagonal elements equal to the square of the nominal radiometric noise shown in Figure 3. Because of this normalization, we do not expect large deviation of the estimated radiometric noise (especially for CrIS band 1 or LWIR) from that shown in Figure 3, because, as mentioned, the figure refers to an assessment of CrIS noise performed in December 2018 [17] with the data recorded at the FSR mode. We do expect possible differences in band 2 and largely in band 3 because of the resampling performed for these two bands. We recall that the original NEDN shown in Figure 3 was obtained at the CrIS full spectral resolution of 0.625 cm${}^{-1}$ and simply rescaled to the nominal sampling of bands 2 and 3, respectively.

The results for two datasets (one corresponding to September 2015 and the second to November 2015) are shown in Figure 10 and compared to the nominal radiometric noise shown in Figure 3. In addition, for this case, the optimal τ selected through the BIC criterion is shown in Table 3. In Figure 10, we see that the two datasets do not show any important differences: the two radiometric noise curves perfectly overlap in each band. As expected, for LWIR, we found a very nice agreement with CrIS noise performance [17] because in this case we use exactly the same sampling of 0.625 cm${}^{-1}$. For the MWIR and SWIR, although we do take into account the factors $\sqrt{2}$ and 2 for MWIR and SWIR, respectively, our results show a slightly better performance. Our results are fully in agreement with the findings shown in [17]. In fact, according to Han et al. [17], the radiometric noise at FSR mode is expected to be larger than those at NSR mode, and we stress that our CrIS spectral radiances have been processed at NSR mode. In addition, we stress that the results shown in Figure 10 were obtained by averaging over the nine CrIS pixels in order to improve statistics.

Finally, also for CrIS, we checked for non-zero off-diagonal terms in the covariance matrix. The correlation matrix for the dataset on 30 September 2015 is shown in Figure 11 for the three CrIS bands. It is seen that, for bands 1 and 2, there is no strong evidence of extra correlation. Therefore, the use of a diagonal covariance matrix is correct. Conversely, for band 3, we do see an excess of correlation, although this is limited to below ±0.3. The same pattern is seen also with the second dataset recorded on 9 November 2015 and not shown here for the sake of brevity. This excess of correlation, mostly seen in band 3, is in agreement with the results shown in [17]. According to Han et al. [17], the correlation is due to the self-apodization correction and increase at high-frequency channels of the short-wave band spectra.

### 3.3. HIRAS

For HIRAS, the covariance matrix used to normalize the data is diagonal with the diagonal elements equal to square of the nominal radiometric noise shown in Figure 4. The sampling interval Δσ is equal to 0.625 cm${}^{-1}$ for each HIRAS band. As for CrIS, because of this normalization, we do not expect large deviation of the estimated radiometric noise from that shown in Figure 4, because, as mentioned, the figure refers to the HIRAS nominal noise obtained directly from the blackbody calibration targets (both warm and cold).

The results for the radiometric noise are summarized in Figure 12 and compared to that shown in Figure 4. The dataset used in the analysis is made up of tropical spectra recorded on 15 July 2019 over the Indian Ocean. For the sake of brevity, no attempt was done to perform an analysis for single pixels. The analysis was also performed with spectra at higher latitudes and we confirmed the same results we show for the tropical belt.

The optimal τ selected by the BIC criterion is again shown in Table 3. In Figure 12, we see that HIRAS noise is better than the nominal one in all bands and specifically in the SWIR band. This result parallels that shown in [9]. Overall, we can say that the noise largely behaves as expected.

As done for IASI-C and CrIS, we now show and discuss the estimated correlation for the three bands. The results are shown in Figure 13.

In Figure 13, at a first glance, we cannot see any long-range correlation pattern as is the case, e.g., for IASI-C; however, a short range correlation develops around the diagonal, which is mostly visible in HIRAS band 3 or SWIR. A signature around the diagonal appears also in bands 1 (LWIR) and 2 (MWIR), but this is fainter than that in band 3. This kind of short range correlation is most likely due to the application of correcting filters to the original interferogram samples. It is likely that the filter acts as a sort of apodization and lowers the variances at the expense of off-diagonal terms. This effect could also explain the results we show in Figure 12 for the SWIR spectral band.

Finally, Table 3 allows us to compare the various values of the optimal τ, selected through the BIC criterion, for the various experiments. We see that for IASI the use of the three bands merged together leads to a larger τ than that selected for CrIS and HIRAS. We think that this effect is because IASI has a better spectral sampling, which, combined to the three bands merged together, increases the signal complexity, hence the larger τ to represent the data.

## 4. Discussion

The PCA methodology developed in [3,4] was used to assess the radiometric characteristics of three major high-spectral-resolution infrared sensors, which have been put in orbit to pursue the objective of improving Numerical Weather Forecasts.

We showed that IASI-C behaves according to what was expected based on the analysis performed during the pre-flight and the commissioning phase, which ended May 2019. The instrument shows correlation patterns, which reach consistent values mostly in the merging of bands 1 and 2. It seems that CNES engineers can now reproduce these complicated correlation patterns and properly take them into account into the observational covariance matrix (e.g., see Figure 2). We also showed that the radiometric performance of the four IASI pixels is quite homogeneous and we could not see any important deviation of one pixel from the others.

In the case of CrIS, we confirmed the excellent radiometric noise performance, which has been reported in previous studies (e.g., [2]). In agreement with Han et al. [17], it seems that, once resampled at 1.25 cm${}^{-1}$ in the MWIR and 2.5 cm${}^{-1}$ in the SWIR, CrIS performed even better than expected on a simple statistical scaling of the radiometric noise. Again, in agreement with Han et al. [17], we evidenced an excess of unexpected correlation in band 3, which is likely the effect of the self-apodization correction.

Finally, for HIRAS, we confirmed that the instrument is behaving as expected. However, once again, we noticed unexpected extra-correlation terms, which tend to be more consistent along the diagonal (short-range correlation). The origin of this correlation, which reaches its maximum in HIRAS band 3, is likely the effect of corrections applied with the algorithms for processing the spectral radiances.

To summarize our analysis, Figure 14 compares the radiometric noise of the three instruments. For CrIS, we used the data provided to us by Han et al. [17] because they apply to the CrIS standard sampling of 0.625 cm${}^{-1}$ and, therefore, can be directly compared to the HIRAS radiometric noise as estimated in this study (see Figure 12). For IASI, we showed values calculated at the present sampling of 0.25 cm${}^{-1}$ (level 1C, apodized data) and those after properly rescaling IASI at CrIS nominal sampling (IASI at CrIS resolution) of 0.625 cm${}^{-1}$ unapodized.

It is seen that CrIs behaves better than IASI and HIRAS in bands 1 and 3. IASI still has a slightly better performance in band 2. Overall, HIRAS compares well with IASI original sampling, but it provides a number of spectral radiances, which is a factor 3.6 smaller.

## Figures and Tables

**Figure 1 sensors-20-01492-f001:**
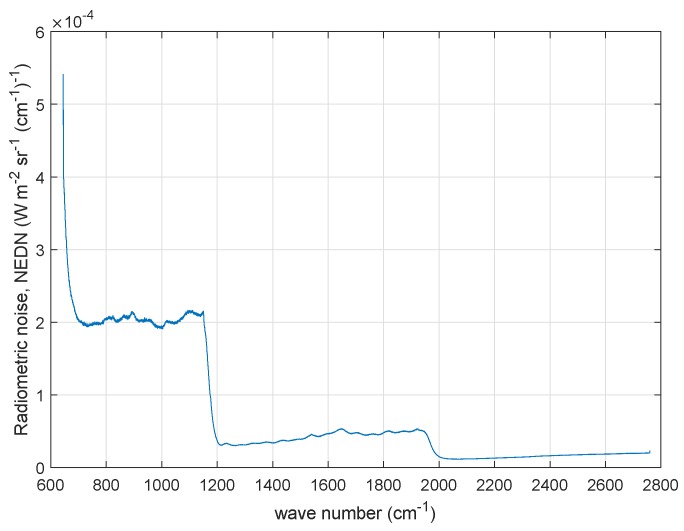
IASI-C radiometric noise in units of NEDN (W-m${}^{-2}$ sr${}^{-1}$ (cm${}^{-1}$)${}^{-1}$). The noise has been averaged over the four IASI pixels or IFOVs.

**Figure 2 sensors-20-01492-f002:**
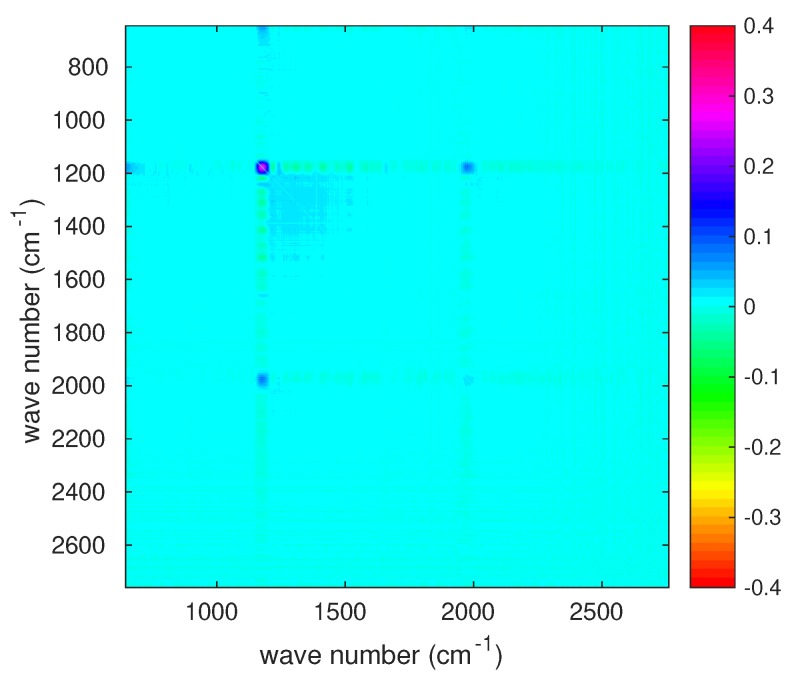
IASI-C $S_{\varepsilon }$ transformed to correlation matrix; the diagonal has been subtracted to better identify correlation structures.

**Figure 3 sensors-20-01492-f003:**
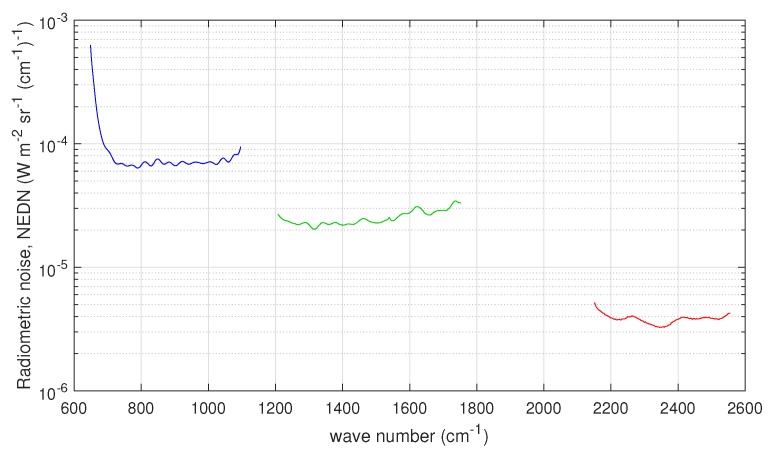
CrIS radiometric noise in units of NEDN (W-m${}^{-2}$ sr${}^{-1}$ (cm${}^{-1}$)${}^{-1}$) for the three bands of the instrument.

**Figure 4 sensors-20-01492-f004:**
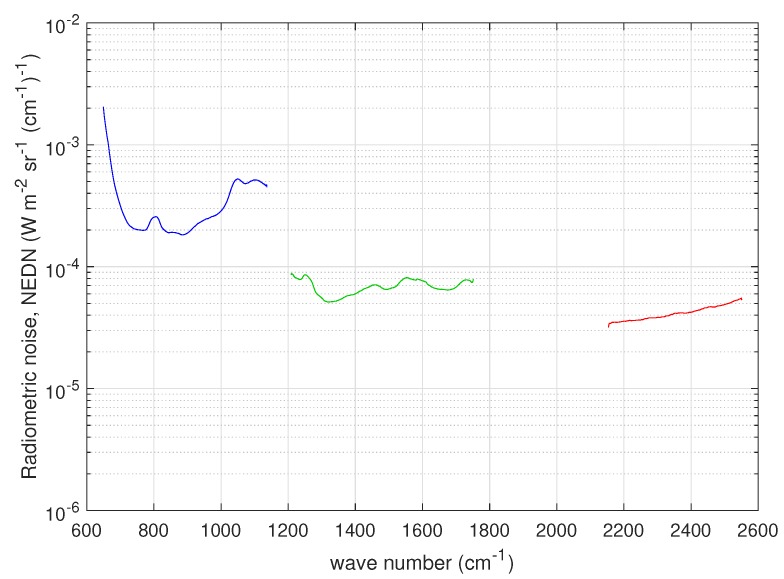
HIRAS radiometric noise in units of NEDN (W-m${}^{-2}$ sr${}^{-1}$ (cm${}^{-1}$)${}^{-1}$) for the three bands of the instrument. The radiometric figures have been obtained by averaging over the four HIRAS pixels and correspond to the FSR sampling of 0.625 cm${}^{-1}$ of the instrument.

**Figure 5 sensors-20-01492-f005:**
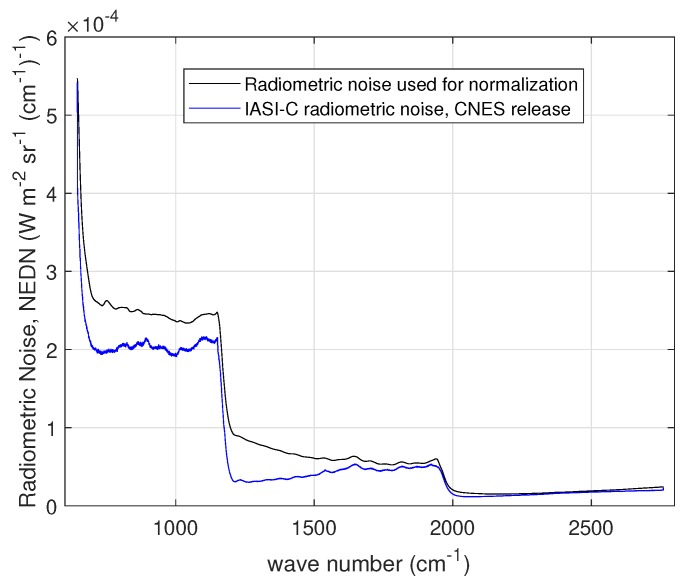
Diagonal of the normalizing matrix ${\tilde{S}}_{\varepsilon }$ used for IASI and comparison with the IASI-C nominal radiometric noise in units of NEDN (W-m${}^{-2}$ sr${}^{-1}$ (cm${}^{-1}$)${}^{-1}$) (also shown in Figure 1).

**Figure 6 sensors-20-01492-f006:**
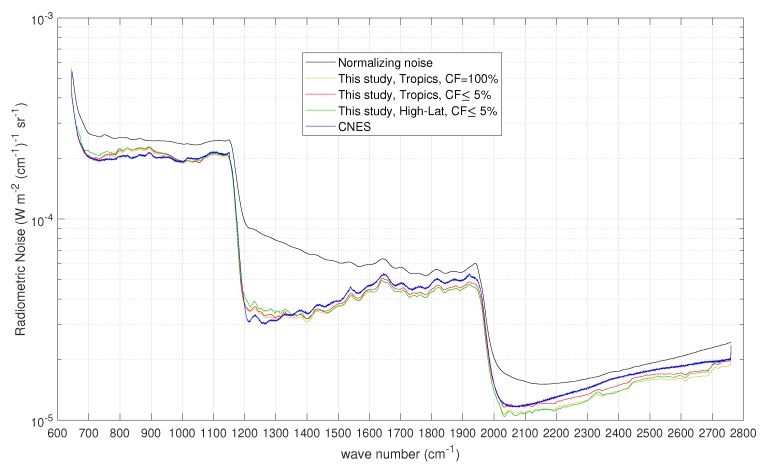
Results for the radiometric noise for the IASI-C instrument. The figure also shows the noise used for normalization and the nominal radiometric noise according to the CNES release [20].

**Figure 7 sensors-20-01492-f007:**
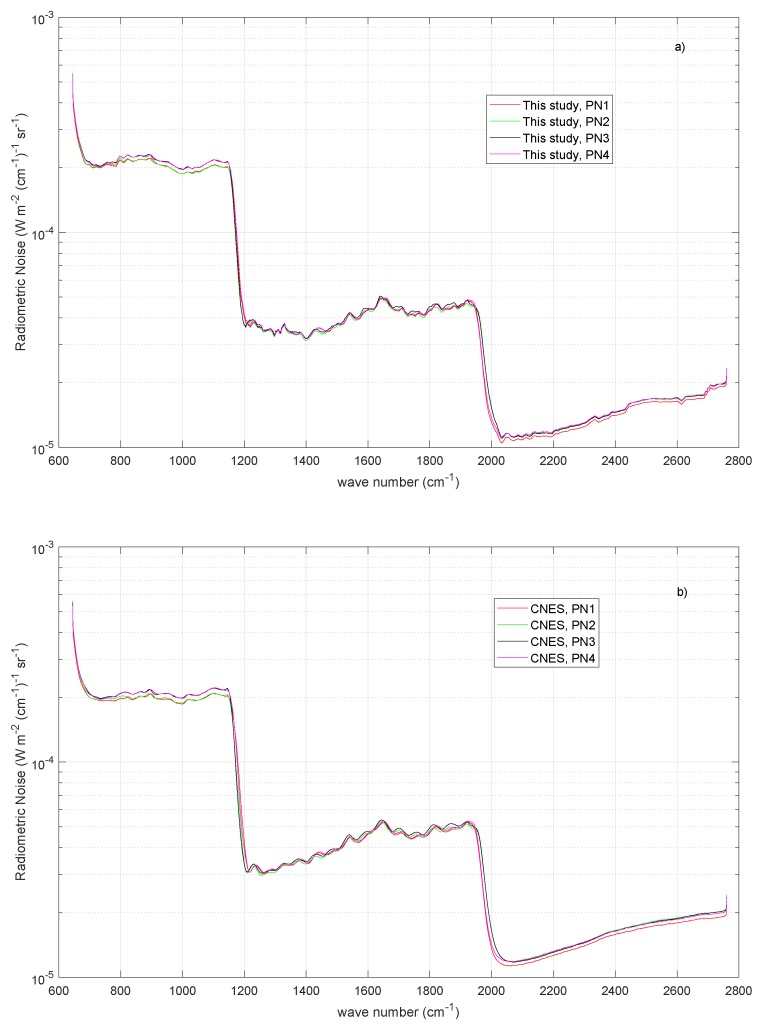
(**a**) Results for the radiometric noise for the IASI-C instrument as a function of the IASI pixel or IFOV; and (**b**) same as (**a**), but now for comparison the CNES release of the nominal radiometric noise is shown.

**Figure 8 sensors-20-01492-f008:**
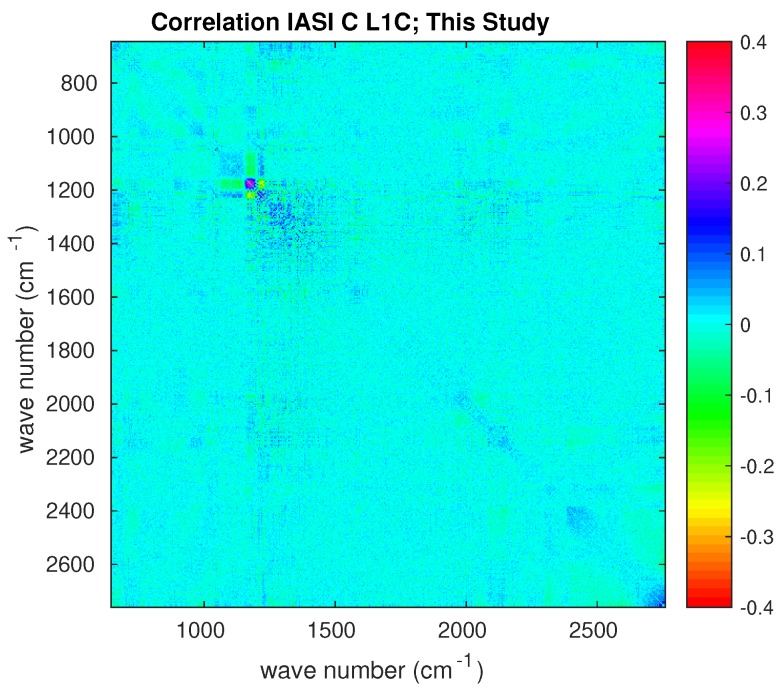
IASI-C $S_{\varepsilon }$ transformed to correlation matrix estimated on the basis of Earth-scene views. The diagonal has been subtracted to allow for a better visual inspection for possible correlation patterns. Results have been averaged over the four IASI pixels.

**Figure 9 sensors-20-01492-f009:**
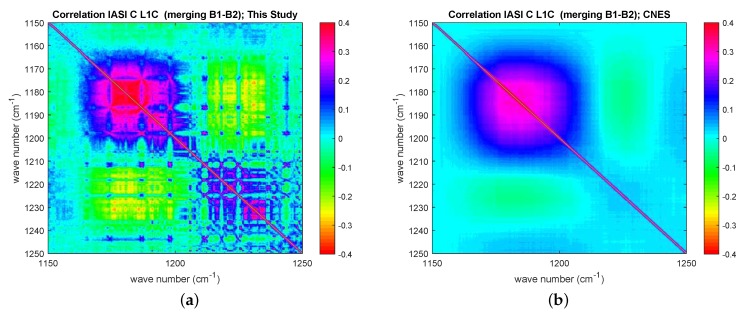
(**a**) IASI-C $S_{\varepsilon }$ transformed to correlation matrix estimated on the basis of Earth-scene views. The figure zooms in the spectral range corresponding to the merging of IASI bands 1 and 2. (**b**) The same as in (**a**), but now the CNES release is shown for comparison. The diagonal has been subtracted to allow for a better visual inspection for possible correlation patterns. Results have been averaged over the four IASI pixels.

**Figure 10 sensors-20-01492-f010:**
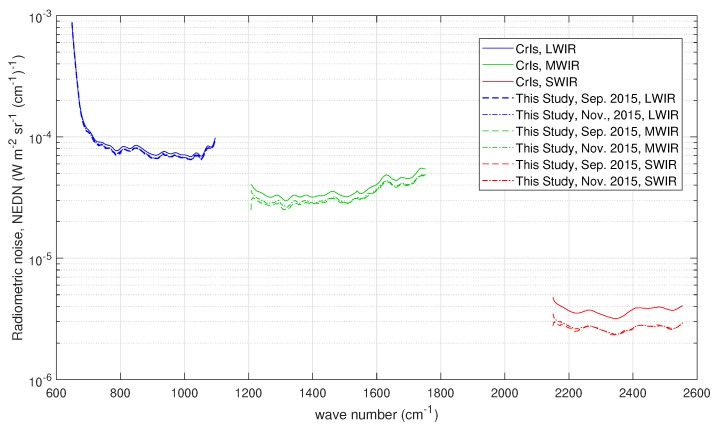
Results for the radiometric noise for the CrIS instrument. The figure also shows the noise used for normalization, which in this case corresponds to the CrIS radiometric noise assessed in [17]. The results have been averaged over the nine CrIS pixels or IFOVS.

**Figure 11 sensors-20-01492-f011:**
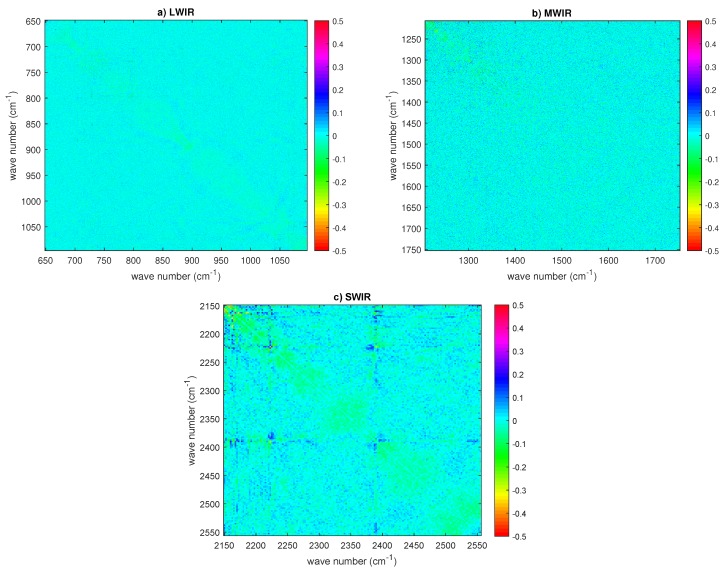
CrIS estimated correlation matrix (data set on 30 September 2015) for: (**a**) LWIR; (**b**) MWIR; and (**c**) SWIR. The sampling is 0.625 (LWIR), 1.25 (MWIR), and 2.5 cm${}^{-1}$ (SWIR). In all correlation matrices, the diagonal has been subtracted to allow for a better visual inspection for possible correlation patterns. The results have been averaged over the nine CrIS pixels or IFOVS.

**Figure 12 sensors-20-01492-f012:**
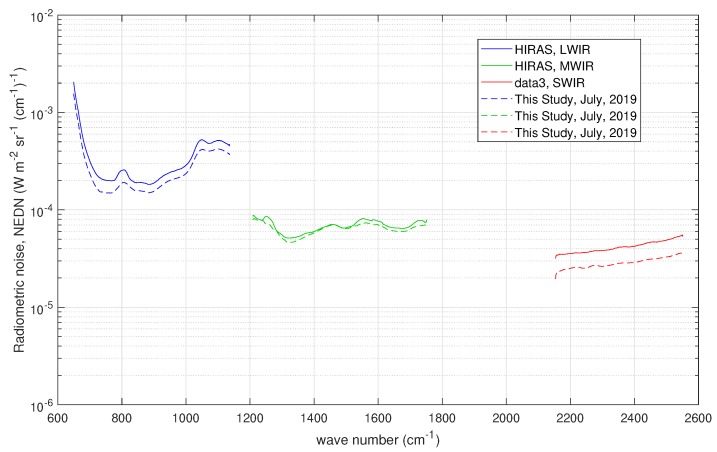
Results for the radiometric noise for the HIRAS instrument. The figure also shows the noise used for normalization, which in this case correspond to the HIRAS nominal radiometric noise shown in Figure 4. Results have been averaged over the four HIRAS pixels.

**Figure 13 sensors-20-01492-f013:**
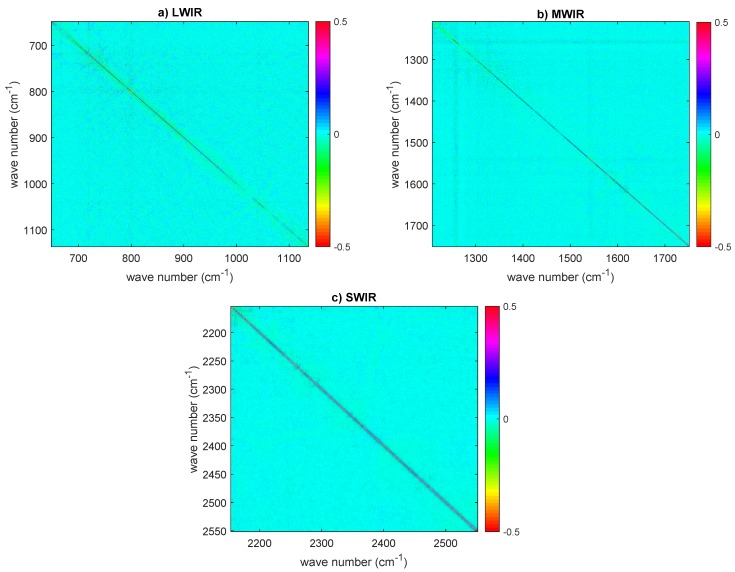
HIRAS estimated correlation matrix (dataset corresponding to the tropical belt and recorded on 8–9 June 2018) for: (**a**) LWIR; (**b**) MWIR; and (**c**) SWIR. The sampling is 0.625 cm${}^{-1}$ in each band. In all correlation matrices, the diagonal has been subtracted to allow for a better visual inspection for possible correlation patterns. Results have been averaged over the four HIRAS pixels.

**Figure 14 sensors-20-01492-f014:**
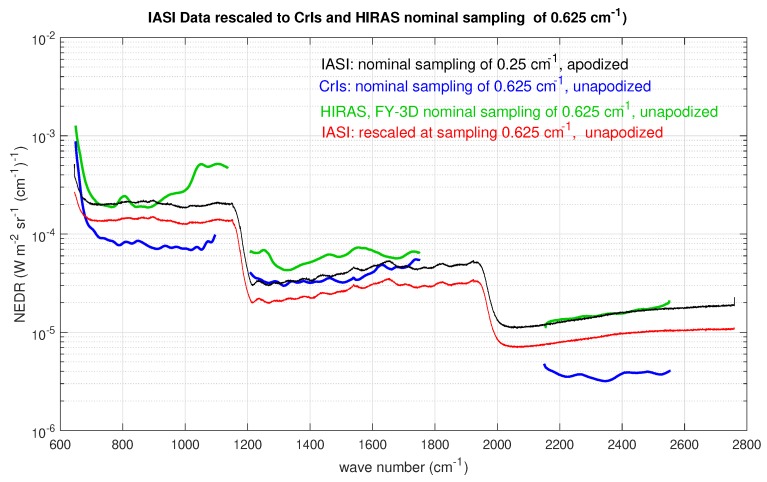
Comparison of IASI, CrIS, and HIRAS radiometric noise. The CrIS data shown in the figure are those provided to us by Han et al. [17], which apply to the CrIS standard sampling of 0.625 cm${}^{-1}$. In this way, they can be directly compared to those of HIRAS. For IASI, we show both the radiometric noise at the original sampling and those properly rescaled at CrIS standard sampling.

**Table 1 sensors-20-01492-t001:** IASI instrument and scanning characteristics.

Characteristic	Value	Units
Spectral Coverage	645 to 2760	cm${}^{-1}$
Number of Bands	3	
Spectral Coverage Band 1	645 to 1210	cm${}^{-1}$
Spectral Coverage Band 2	1210 to 2000	cm${}^{-1}$
Spectral Coverage Band 3	2000 to 2760	cm${}^{-1}$
Scan Type	Step and Stare	
Scan Rate	8	s
Stare Interval	151	ms
Step Interval	8/37	s
Pixels for Field of Regard (FOR)	2×2	
Swath	±48.333	degrees
Swath width	±1100	km
Number of FOR per Swath width	30	
Single Pixel or Instantaneous Field of View (IFOV) shape at nadir	circular	
IFOV size at nadir	12	km
IFOV size at edge of scan line across track	39	km
IFOV size at edge of scan line along track	20	km

**Table 2 sensors-20-01492-t002:** CrIS instrument and scanning characteristics.

Characteristic	Value	Units
Spectral Coverage	648.75 to 2555	cm${}^{-1}$
Number of Bands	3	
Spectral Coverage Band 1	648.75 to 1096.25	cm${}^{-1}$
Spectral Coverage Band 2	1207.50 to 1752.50	cm${}^{-1}$
Spectral Coverage Band 3	2150 to 2555	cm${}^{-1}$
Scan Type	Step and Stare	
Scan Rate	8	s
Pixels for Field of Regard (FOR)	3×3	
Swath	±50	degrees
Swath width	±1100	km
Number of FOR per Swath width	30	
Single Pixel or Instantaneous Field of View (IFOV) shape at nadir	circular	
IFOV size at nadir	14	km

**Table 3 sensors-20-01492-t003:** Number of PC scores used in the analysis for the different instruments and datasets. For IASI, all bands are merged together.

Data Set		τ			No. Spectra
		Band 1	Band 2	Band 3	
			**IASI**		
Tropical CF = 100%			298		14,321
Tropical CF ≤ 5%			185		9324
High Latitude CF ≤ 5%			276		11,230
Tropical all sky	Pixel 1		253		10,240
	Pixel 2		288		10,240
	Pixel 3		230		10,240
	Pixel 4		225		10,240
			**CrIS**		
September 2015		95	75	20	10,080
November 2015		89	78	22	9360
			**HIRAS**		
July 2019		93	83	34	8320

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
