# Peer review of "Characterization of the Observational Covariance Matrix of Hyper-Spectral Infrared Satellite Sensors Directly from Measured Earth Views"

_sensors, 2020, doi:10.3390/s20051492_

Round 1

Reviewer 1 Report

The authors apply a Principal Component Analysis to characterise the noise of three infrared satellite sensors. Their method is a useful complement to the standard analysis. I found no major flaws in their manuscript. One thing, however, that I suggest to add to the article, is a discussion of the uncertainty of the radiometric noise values the authors present. They compare their own noise values to those obtained in previous studies, but it is hard to judge how significant the differences are. Is it fair to say that in this paper the only thing new is the detection of extra correlation among channels in the short wave regions of the spectra?

Suggestions for minor changes:

Figure 2: The colour scale is badly chosen - only a tiny fraction of the available colours appears in the image.

Line 199: Explain the abbreviation "SVD".

Line 223: Wouldn't it be good to include the very first CNES release in the list of References?

Figure 6: It is difficult to distinguish the pink and the red line, being very close to each other and looking almost the same.

References: The use of italics varies from one reference to the next, and there are several typing errors (also elsewhere in the manuscript). 

Author Response

Please track in red our reply to Reviewer 1

The authors apply a Principal Component Analysis to characterize the noise of three infrared satellite sensors. Their method is a useful complement to the standard analysis. I found no major flaws in their manuscript.

First of all we want to thank the reviewer for judging our manuscript useful after carefully reading it. We thank the reviewer also for the suggestions and corrections that contribute to make it clearer for the readers.

One thing, however, that I suggest to add to the article, is a discussion of the uncertainty of the radiometric noise values the authors present. They compare their own noise values to those obtained in previous studies, but it is hard to judge how significant the differences are. Is it fair to say that in this paper the only thing new is the detection of extra correlation among channels in the short wave regions of the spectra?

We thank the reviewer for this comment, which allows us to better qualify the objective of the work. For the discussion about uncertainty it has to be said that for a covariance statistics such as that of Eq. (8), under the assumptions of Gaussian noise, the uncertainty is given by the Wishart distribution (e.g., Anderson, 2009), which yield the statistical sample probability distribution of covariance estimators. If  sij  denotes the true values of the matrix Sε , then the variance of the estimate Sε(i,j) is given by

Var(Sε(i,j))=1/N *(sij2+sii sjj)

which for i=j (variances), becomes

Var(Sε(i,j))=2/N *sii2

That is the variance of the variance estimator is proportional to square of the true variance and inverse proportional to the statistical weight N. Therefore, the uncertainty of our estimate of the radiometric noise is, in percentage, of the order of sqrt(2/N), which for the analysis presented here, considering that N≅104, yields a value of the order of ≈1-2%. We added a brief paragraph in section 2.2 to explain and discuss error uncertainty. By the way, our results are just not a mere comparison with previous ones. The result for correlation in band 3 applies to CrIS, for IASI and HIRAS the correlation is affecting all bands. Previous results for CrIS, used in this paper, have been obtained in the FSR mode, whereas our results apply to the NSR mode. For the three instruments, the radiometric noise does not show not show a strong dependence on the scene.

Suggestions for minor changes:

Figure 2: The colour scale is badly chosen - only a tiny fraction of the available colours appears in the image.

Done: Thanks. According this suggestion we change also Figure 8.

Line 199: Explain the abbreviation "SVD".

Done: Thanks, we add Singular Value Decomposition before the acronym.

Line 223: Wouldn't it be good to include the very first CNES release in the list of References?

Done: The very first CNES release has not been published until the paper by Serio et al 2018 (Ref. 4 in the paper) . We have also made the reference to the CNES report “E. Jacquette and J. Chinaud, IASI Quarterly Performance Report from 2012/09/01 to 2012/11/30 by IASI TEC (Technical Expertise Center) for IASI PFM-R on METOP A (CNES, 2013)”.

Figure 6: It is difficult to distinguish the pink and the red line, being very close to each other and looking almost the same.

Done: Thanks, we used dark yellow line.

References: The use of italics varies from one reference to the next, and there are several typing errors (also elsewhere in the manuscript).

Done: Thanks, we homogenized references style, we also correct several typing errors.

Reviewer 2 Report

The manuscript deals with an estimate of the observational covariance matrix from operational radiance measured from satellites for 3 sensors (IASI, CrIS, HIRAS).

The subject is interesting to the community. The paper is clearly written (even though with several misprints), the results are original and significant. Therefore the manuscript deserves publication in the Journal

I'm asking to make the following corrections:

a) image of the covariance matrix is the key result of the manuscript (together with plot of the diagonal). However chromatic appearance is not satisfactory for Figure 2. Indeed no structure is visible, except barely one or two points in a color with a bad contrast with background. The problem is the color scale that ranges from -1 to 1, whereas the actual range of correlation seems to be limited to around 0.4 (in absolute value), once the diagonal part has been removed. This also applies to Figure 8. On the contrary correlation matrices of Figure 9 are represented with a color range up to 0.4 (in absolute value) that fully exploits the chromatic scale. If it is confirmed that correlation of Figures 2 and 8 is less than 0.4 (in absolute value), authors should use the same range [-0.4,4] for them.

b) authors need a "priori" observational covariance matrix ($\tilde S_\epsilon$) for their method, and they choose a rough old estimate from CNES for IASI-A and similar estimates for CrIS and HIRAS. After elaboration, they finally get the estimate $\hat S_\epsilon$ (Eq. 9). What about to iterate the procedure substituting $\tilde S_\epsilon$ with the latest $\hat S_\epsilon$? At convergence they should get that the two matrices are equal (and $S$ of Eq. 4 identical). This could make their method more robust with respect to the choice of$\tilde S_\epsilon$. Can authors comment?

c) report k (number of PC) selected by BIC for the various experiments

d) Correct the following misprints:

  • l. 36: dependencies instead of dependency
  • l. 45: "into an orthogonal basis" instead of "in an orthogonal basis"
  • l. 79: onboard instead of aboard
  • l. 91: Data instead of data
  • l. 93: onboard instead on on board
  • l. 131: "can do" is not an admissible construct, please rephrase
  • l. 143: pattern instead of patter
  • l. 146: correct 10196.25
  • l. 161: bands instead of band
  • l. 164: put a dot at the end of the sentence
  • l. 195: covariance matrix instead of matrix
  • l. 199: "Let $X$ be" instead of "Let $X$"
  • l. 199: "diagonal values $\lambda_j$, $j=1,\ldots,d$" instead of "elements $\lambda_j$"
  • l. 214: as instead of ass
  • l. 219: please rephrase "findings we have found"
  • l. 229: Figure 5: Radiometric instead of Rdaiometric
  • l. 230: close left parenthesis
  • l. 235 (and everywhere): use \le instead of <= (also in Figure 6)
  • l. 244: remove shown
  • l. 245: rephrase "in the case that used for..."
  • l. 256: close left parenthesis
  • l. 258: pixels instead of pixel
  • l. 264: does instead of do
  • l. 266: shows instead of show
  • l. 272: the square instead of square
  • l. 276: recall instead of remember
  • l. 279: remove the
  • l. 284: remove of
  • l. 303: "from that" instead of "of that"
  • l. 308: Indian Ocean instead of Ocean Indian
  • l. 319: remove "a"
  • l. 338: Figure 14: red and magenta colors are not well distinguishable; please use a different color for one of the twos
  • l. 427: I think should be removed

Author Response

Please track in red our reply to Reviewer 2

The manuscript deals with an estimate of the observational covariance matrix from operational radiance measured from satellites for 3 sensors (IASI, CrIS, HIRAS).

The subject is interesting to the community. The paper is clearly written (even though with several misprints), the results are original and significant. Therefore the manuscript deserves publication in the Journal

First of all we want to thank the reviewer for judging our manuscript original, significant and deserving publication after carefully reading it. We thank the reviewer also for the precise suggestions and corrections that contribute to make it clearer for the readers.

I'm asking to make the following corrections:

a) image of the covariance matrix is the key result of the manuscript (together with plot of the diagonal). However chromatic appearance is not satisfactory for Figure 2. Indeed no structure is visible, except barely one or two points in a color with a bad contrast with background. The problem is the color scale that ranges from -1 to 1, whereas the actual range of correlation seems to be limited to around 0.4 (in absolute value), once the diagonal part has been removed. This also applies to Figure 8. On the contrary correlation matrices of Figure 9 are represented with a color range up to 0.4 (in absolute value) that fully exploits the chromatic scale. If it is confirmed that correlation of Figures 2 and 8 is less than 0.4 (in absolute value), authors should use the same range [-0.4,4] for them.

Done: Thanks.

b) authors need a "priori" observational covariance matrix ($\tilde S_\epsilon$) for their method, and they choose a rough old estimate from CNES for IASI-A and similar estimates for CrIS and HIRAS. After elaboration, they finally get the estimate $\hat S_\epsilon$ (Eq. 9). What about to iterate the procedure substituting $\tilde S_\epsilon$ with the latest $\hat S_\epsilon$? At convergence they should get that the two matrices are equal (and $S$ of Eq. 4 identical). This could make their method more robust with respect to the choice of$\tilde S_\epsilon$. Can authors comment?

As shown in Ref. (4), our methodology is linear and is not heavily dependent on the choice of the a priori estimate. Thus, we do not expect any important improvement by iterating on the a priori. In other words, since the problem is linear, we expect to do the job in one iteration. We a bit rephrased the sentence on line 245 to better convey this information.

c) report k (number of PC) selected by BIC for the various experiments

Done: at the end of Section 3 we added  a Table  with the number of PC for all the data sets used in this work. Thanks

d) Correct the following misprints:

  • l. 36: dependencies instead of dependency. Done: Thanks
  • l. 45: "into an orthogonal basis" instead of "in an orthogonal basis". Done: Thanks
  • l. 79: onboard instead of aboard. Done: Thanks
  • l. 91: Data instead of data. Done: Thanks
  • l. 93: onboard instead on on board. Done: Thanks
  • l. 131: "can do" is not an admissible construct, please rephrase. Done: we rephrased in this way “Slight variations among pixels are expected to exist”. Thanks
  • l. 143: pattern instead of patter. Done: Thanks
  • l. 146: correct 10196.25. Done: we correct in 1096.25. Thanks
  • l. 161: bands instead of band. Done: Thanks
  • l. 164: put a dot at the end of the sentence. Done: Thanks.
  • l. 195: covariance matrix instead of matrix. Done: Thanks
  • l. 199: "Let $X$ be" instead of "Let $X$". Done: Thanks
  • l. 199: "diagonal values $\lambda_j$, $j=1,\ldots,d$" instead of "elements $\lambda_j$". Done: Thanks.
  • l. 214: as instead of ass. Ops, we’re sorry! Done: Thanks.
  • l. 219: please rephrase "findings we have found". Done: we rephrased in this way “results we have found”. Thanks
  • l. 229: Figure 5: Radiometric instead of Rdaiometric. Done: Thanks.
  • l. 230: close left parenthesis. Done: Thanks.
  • l. 235 (and everywhere): use \le instead of <= (also in Figure 6). Done: Thanks.
  • l. 244: remove shown. Done: Thanks.
  • l. 245: rephrase "in the case that used for...". Done: We rephased this way the whole sentence also to address the remark b) done by the reviewer: “in the case the covariance matrix used for normalization is far apart from the true noise, which is in this analysis is assumed to be that released by CNES [20]. In passing, Fig. 6 also shows that problem is largely linear and converges in one step (for further details we refer the reader to [4])”.
  • l. 256: close left parenthesis. Done: Thanks.
  • l. 258: pixels instead of pixel. Done: Thanks.
  • l. 264: does instead of do. Done: Thanks.
  • l. 266: shows instead of show. Done: Thanks.
  • l. 272: the square instead of square. Done: Thanks.
  • l. 276: recall instead of remember. Done: Thanks.
  • l. 279: remove the. Done: Thanks.
  • l. 284: remove of. Done: Thanks.
  • l. 303: "from that" instead of "of that". Done: Thanks.
  • l. 308: Indian Ocean instead of Ocean Indian. Done: Thanks.
  • l. 319: remove "a". Done: Thanks.
  • l. 338: Figure 14: red and magenta colors are not well distinguishable; please use a different color for one of the twos. Done: Thanks, we used green line for HIRAS.
  • l. 427: I think should be removed. Removed: Thanks.